# Prompting Scientific Names for Zero-Shot Species Recognition

**Shubham Parashar[1], Zhiqiu Lin[2], Yanan Li[3,*], Shu Kong[4,5,1,*]**

[1]Texas A&M University, [2]Carnegie Mellon University, [3]Zhejiang Lab,
[4]Institute of Collaborative Innovation, [5]University of Macau

{shubhamprshr, shu}@tamu.edu, zhiqiulin@cmu.edu, liyn@zhejianglab.com, skong@um.edu.mo

## Abstract

Trained on web-scale image-text pairs, Vision-Language Models (VLMs) such as CLIP (Radford et al., 2021) can recognize images of common objects in a zero-shot fashion. However, it is underexplored how to use CLIP for zero-shot recognition of highly specialized concepts, e.g., species of birds, plants, and animals, for which their scientific names are written in Latin or Greek. Indeed, CLIP performs poorly for zero-shot species recognition with prompts that use scientific names, e.g., "a photo of `Lepus Timidus`" (which is a scientific name in Latin). This is because such names are usually not included in CLIP's training set. To improve performance, prior works propose to use large-language models (LLMs) to generate descriptions (e.g., of species color and shape) and additionally use them in prompts. We find that they bring only marginal gains. Differently, we are motivated to translate scientific names (e.g., `Lepus Timidus`) to common English names (e.g., `mountain hare`) and use such in the prompts. We find that common names are more likely to be included in CLIP's training set, and prompting them achieves 2∼5 times higher accuracy on benchmarking datasets of fine-grained species recognition.

## 1 Introduction

Trained on large-scale image-text data, vision-language models (VLMs) such as CLIP (Radford et al., 2021; Ilharco et al., 2021) and ALIGN (Jia et al., 2021) can recognize images of common objects in a zero-shot fashion, i.e., without further finetuning. Such success is typically reported on standard benchmarks such as ImageNet which contains 1,000 common classes, e.g., CLIP achieves 76.2% zero-shot accuracy vs. 11.5% by Li et al. (2017) prior to CLIP. We are motivated to use CLIP for zero-shot recognition of highly specialized concepts, e.g., species of plants, birds, and animals.

---

* Corresponding to Yanan Li and Shu Kong.

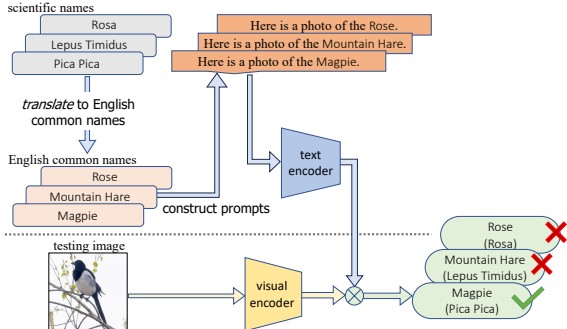

Figure 1: We propose a rather simple method that boosts performance for zero-shot species recognition. It translates scientific names (written in Latin or Greek) to common English names, and use the latter in prompts. While using large language models (LLMs) can translate scientific names, they can fail for many species (Fig. 2). Instead, common names can be found in other publicly available sources such as online collections and museums. By replacing scientific names with common names, simple prompt method achieves 2-5 times better zero-shot species recognition accuracy on challenging benchmarks (Table 1).

In other words, we study the problem of *zero-shot species recognition using VLMs*.

**Motivation**. Zero-shot species recognition is practically meaningful in various applications, e.g., for education and ecological and biodiversity research, which desire automated recognition of species (Stork et al., 2021; Rodr'iguez et al., 2022). Building species recognition systems can hardly rely on traditional supervised learning methods which require a large set of labeled data, because labeling images with species names demands domain expertise which is expensive to obtain. Therefore, we propose to leverage pretrained VLMs (e.g., CLIP) for zero-shot species recognition.

**Technical Insights**. Species have scientific names written in Latin or Greek. Directly using them in prompts (e.g., "a photo of `Ponana Citrina`") does not allow OpenCLIP (an open-source CLIP) (Ilharco et al., 2021) to perform well (Fig. 3). This is not quite surprising, because, for the first time, we find that OpenCLIP's training set LAION400M (Schuhmann et al., 2021) does not contain scientific names of many species . Per-

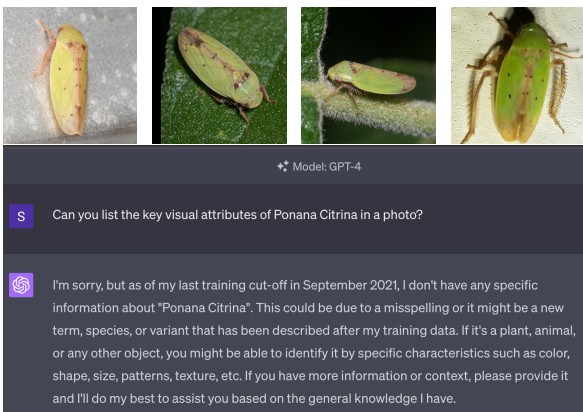

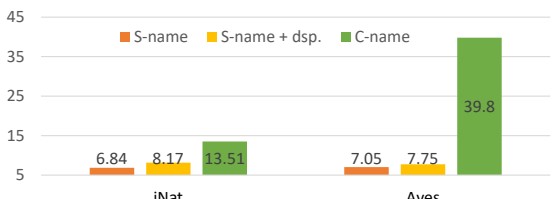

Figure 2: Top-row: four images from the species Ponana Citrina; bottom-row: GPT4 fails to answer questions related to this species, demonstrating a limitation of LLMs w.r.t understanding scientific names.

Figure 3: We compare methods (1) vanilla prompt (Radford et al., 2021) using scientific names which are written in Latin or Greek (*S-name*), (2) S-name plus descriptions (Menon and Vondrick, 2022), and (3) our prompt method using common names (*C-name*). Results are top-1 accuracy on two datasets (iNat and Aves detailed in Sec. 4.1). Clearly, using scientific names in prompts yields poor zero-shot species recognition performance because pretrained VLMs do not necessarily see scientific names. Additionally using descriptions improves marginally. In contrast, simply translating them to common English names significantly boost performance.

## 2 Related Work

**Zero-shot recognition**. Previous methods exploit auxiliary semantic information such as attributes (Lampert et al., 2013) or distributed word embedding (Elhoseiny et al., 2017) of class names and learn a mapping between them for knowledge transfer from base to novel classes (Xian et al., 2018). Recently, zero-shot VLMs pre-trained on large-scale image-text corpus such as CLIP (Radford et al., 2021) achieve state-of-the-art performance on a wide range of image recognition benchmarks. Specifically, these models do not require fine-tuning on downstream datasets so long as the class names are provided. In this work, we use OpenCLIP (Ilharco et al., 2021) as the zero-shot VLM for recognition but focus on more specialized and underexplored tasks, i.e., species recognition.

**Prompting approaches**. To use VLMs for zero-shot recognition, it is crucial to design prompts. Initially, prompts are manually engineered (Radford et al., 2021) which requires trials and errors. Some methods propose to learn prompt templates (Zhou et al., 2022; Bulat and Tzimiropoulos, 2022), which require labeled data and are hard to optimize. Some recent work proposes to exploit LLMs (Bubeck et al., 2023; Touvron et al., 2023) to generate prompts for given concepts (Menon and Vondrick, 2022; Pratt et al., 2023). In this work, we focus on zero-shot species recognition, for which LLMs might fail to provide useful information to help construct prompts (Fig. 2). Further, different from existing prompting approaches, we propose a rather simple method by using species common names (translated from their scientific names). To the best of our knowledge, ours is the first that translates scientific names to common names in prompts for zero-shot recognition.

haps surprisingly, the recent large-language model (LLM) GPT4 can also fail to answer questions related to species scientific names (Fig. 2), such as "can you describe the appearance of Ponana Citrina?". Inspired by Menon and Vondrick (2022), to improve zero-shot recognition performance, we additionally use descriptions in prompts, hoping they provide useful contextual information. However, it only marginally improves performance. On the other hand, although scientific names are not frequently included in OpenCLIP's training set, their corresponding English common names are. Therefore, we are motivated to translate scientific names to common ones and use the latter in prompts (Fig. 1). Our embarrassingly simple method significantly boosts performance by 2∼5 times! To note, while LLMs can fail to understand scientific names[1] and so provide common names, common names can be found in other sources such as online collections and museums.

**Contributions** are two-fold.

1. We study the underexplored problem of zero-shot species recognition using VLMs. We confirm that current prompting methods yield poor performance and find that the culprit is the scientific names written in Latin or Greek, most of which are not in VLM's training set.

2. We propose an embarrassingly simple method that translates scientific names to common names, boosting zero-shot species recognition accuracy by 2∼5 times.

---

[1]It is common that GPT4 can fail to answer human's questions, but its web browsing mode mitigates this by retrieving information on the fly from existing (professional) websites.

## 3 Methods

We start with the vanilla prompts that have been used in the contemporary literature of zero-shot recognition (Radford et al., 2021).

**Vanilla Prompt** uses the prompt template "a photo of `Lepus Timidus`" to get a similarity measure for a given input image. Note that for this method, we use species *scientific names* (written in Latin or Greek) in the prompts, hence call this method *S-name*. Concretely, it feeds prompts to CLIP's text encoder for the text features, uses them to compute cosine similarities to the visual feature computed by the visual encoder for the given image. CLIP tends to produce a high similarity for the correct prompt (corresponding to ground-truth class name), hence achieving zero-shot recognition.

**Prompt with additional descriptions** enriches prompts with descriptions (e.g., shape, color, and size). These descriptions provide contextual information for the targeted concepts and help zero-shot recognition (Menon and Vondrick, 2022). Recent work proposes to generate such descriptions using LLMs (Pratt et al., 2023). Although using LLMs does not always work on specific species (Fig. 2), we follow this practice to generate descriptions using GPT4. When GPT4 fails to provide useful descriptions for any species, we use the simple prompts without any descriptions by default. Note using additional descriptions apply to both vanilla prompt and methods below.

**Prompt with common names** translated from scientific names (*C-name*). We investigate OpenCLIP's training set, i.e., LAION400M (Schuhmann et al., 2021), and find that it does not sufficiently cover scientific names (e.g., 468 out of 810 species in the semi-iNat dataset have their scientific names in LAINON400M). Instead, many species have common names that are in LAION400M. Therefore, we turn to external resources (e.g., online collections and Wikipedia) to translate scientific names (written in Latin or Greek) to common English names and use the latter in prompts. For the species that do not have any common English names (only a few in the benchmarking datasets), we use their original scientific names by default.

**Prompt with more frequent names** (*F-name*) between common and scientific names. We hypothesize that it is important to use the texts that are more frequently encountered by OpenCLIP during training. Therefore, we further analyze LAION400M (Schuhmann et al., 2021) to obtain

Table 1: Benchmarking results of zero-shot species recognition on the iNat and Aves datasets. We use two VLMs: OpenCLIP ViT-B/32 and ViT-L/14 (which is a bigger model). Clearly, using a bigger model produces higher accuracy across methods. The *Vanilla* method that prompts scientific names (S-name) yields quite low accuracy; additionally using descriptions improves accuracy slightly. In contrast, simply prompting common names (C-name) by translating scientific names, we boost performance by 2X on iNat and 5X on Aves, justifying our hypothesis that OpenCLIP might not be versed in understanding scientific names (because its training texts do not contain such). When using common names, additionally exploiting descriptions does not necessarily improve performance. We conjecture that the common names plus descriptions are not frequently seen by OpenCLIP during training. Furthermore, our final method F-name, which uses the more frequent name between scientific and common names for each species, is among the best methods across methods, datasets, and OpenCLIP models.

| VLM | Prompt Method | iNat (810-way) | Aves (200-way) |
|---|---|---|---|
| ViT-B/32 | S-name (vanilla) | 6.84% | 7.05% |
| | + descriptions | 8.17% | 7.75% |
| | C-name (ours) | 13.51% | 39.80% |
| | + descriptions | 14.42% | 39.75% |
| | F-name (ours) | 13.88% | **39.95%** |
| | + descriptions | **14.47%** | 39.40% |
| ViT-L/14 | S-name (vanilla) | 9.21% | 11.10% |
| | + descriptions | 10.15% | 12.50% |
| | C-name (ours) | 20.17% | 59.00% |
| | + descriptions | 20.04% | **59.90%** |
| | F-name (ours) | **20.32%** | 58.45% |
| | + descriptions | 20.03% | 59.10% |

frequency counts of both scientific and common names for all the species of interest and use the more frequent one in the prompt for the corresponding species.

## 4 Experiments

We conduct extensive experiments to validate our methods. We start with implementation details, datasets and the evaluation metric, and then analyze benchmarking results with in-depth analyses.

### 4.1 Setup

**Implementations**. We implement all the methods in PyTorch on a single NVIDIA A100 GPU. We use the open-source OpenCLIP (i.e., the ViT-B/32 and ViT-L/14 models) (Ilharco et al., 2021).

**Datasets**. We use four datasets that require species-level recognition. While they were initially curated for fine-grained classification via supervised learning, we repurpose them for zero-shot species recognition, i.e., using their validation sets for benchmarking.

- The semi-iNaturalist (iNat) (Su and Maji, 2021b) contains 810 species, covering a broad spectrum of organisms including mammals, plants, birds, insects, fungi, etc.

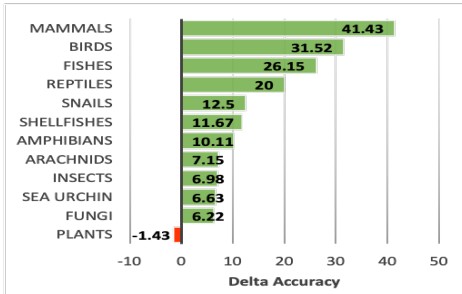

Figure 4: Compared with using scientific names in prompts, using common names achieves significantly higher zero-shot accuracy on all organism types except `plants`. We hypothesize that scientific names of `plants` are more likely to be included in OpenCLIP's training set (LAION400M (Schuhmann et al., 2021)), such that OpenCLIP performs well when directly prompting scientific names. We investigate this by matching the scientific names and LAION400M texts, and find LAION400M contains 65% plant species' scientific names of iNat. To confirm this further, we use the Flowers102 dataset for plant species recognition (Table 2).

- The semi-Aves (Aves) (Su et al., 2021) is curated for 200 bird species recognition.
- The Flowers102 by Nilsback and Zisserman (2008) is a popular fine-grained classification dataset that contains 102 flower types.
- The CUB-200-2011 Bird (CUB200) by Wah et al. (2011) is a popular fine-grained classification dataset that contains 200 bird species.[2]

**Metric**. We use the top-1 accuracy averaged over $K$ per-class accuracy as the metric, where $K$ is the total number of classes.

## 4.2 Results

Table 1 lists benchmarking results. Please refer to the caption for detailed conclusions. We reiterate three salient conclusions. First, using common names in prompts performs significantly better than scientific names. It is worth noting that, on Aves, simply using common names in prompts achieves 59.0% zero-shot recognition accuracy (59.9%), which is even higher than the fully-supervised method (56.6%) that pretrains on ImageNet Su et al. (2021)! Second, additionally using descriptions, although paired with common/scientific names, are not always beneficial, as shown by results on Aves; it can even decrease results (e.g., from 20.32% to 19.93% with the F-name method and ViT-L/14). We conjecture the reason is that OpenCLIP was not trained on texts (similar to prompts) containing species names plus descriptions. Third, F-name, the method that uses the more frequent name between scientific and common names in

---

[2]CUB200 and Flowers102 provide common names. We translate them to scientific names for the vanilla method in experiments.

Table 2: Further analysis on Flowers102 (for plant recognition) and CUB200 (for bird species recognition) datasets. Breakdown results in Fig. 4 show that using common names in prompts improves significantly on organism types such as birds but induces a decrease on plants. Therefore, we use the two more datasets to confirm our hypothesis: OpenCLIP has already seen most of plant species names (different from birds) and hence perform well by prompting scientific names. Indeed, all the prompt methods perform well on Flowers102, although our F-name method performs the best with the bigger model ViT-L/14. Differently, on CUB200, S-name that uses scientific names does not perform well (6.08% with ViT-B/32), but C-name (which uses common names in prompts) boosts accuracy to 59.42% (with ViT-B/32).

| VLM | Prompt Method | Flowers102 (102-way) | CUB200 (200-way) |
|---|---|---|---|
| ViT-B/32 | S-name (vanilla) | 66.43% | 6.08% |
| | + descriptions | **68.67**% | 7.61% |
| | C-name (ours) | 61.89% | 56.01% |
| | + descriptions | 63.33% | 58.44% |
| | F-name (ours) | 68.35% | 58.35% |
| | + descriptions | 66.91% | **59.42**% |
| ViT-L/14 | S-name (vanilla) | 77.28% | 6.92% |
| | + descriptions | 78.53% | 10.44% |
| | C-name (ours) | 73.49% | 76.27% |
| | + descriptions | 74.67% | 76.30% |
| | F-name (ours) | **78.95**% | 76.70% |
| | + descriptions | **78.95**% | **76.87**% |

LAION400M, generally performs the best across datasets and VLM models.

## 4.3 Analysis

Focusing on iNat which contains multiple organism types, we show performance gain of using common names vs. scientific names for each type in Fig. 3. Using common names performs much better on all organisms (e.g., `birds`) except `plants`. This motivates us to investigate the reason using the Flower102 (for plant species recognition) and CUB200 (for bird recognition) datasets. Table 2 lists the results and the caption details conclusions.

## 5 Conclusions

We study the problem of zero-shot species recognition using pretrained vision-language models (VLMs). For this problem, the typical prompt method that uses the species scientific names "by default" performs poorly, and additionally using descriptions in prompts improves only marginally. We find that scientific names written in Latin or Greek are not frequently seen in VLMs' training set, explaining why they do not work well. Yet, species English common names are more likely to be seen in VLMs' training set. Hence, we propose a rather simple method that translates scientific names to common English names, and use the latter in prompts. This method significantly boosts accuracy by 2-5 times on three benchmarking datasets.

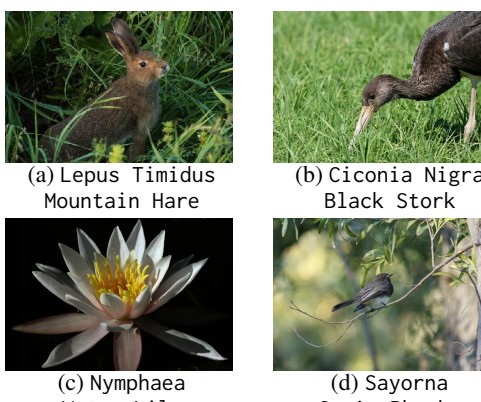

| | | |
|---|---|---|
| (a) Lepus Timidus Mountain Hare | | (b) Ciconia Nigra Black Stork |
| (c) Nymphaea Water Lily | | (d) Sayorna Say's Phoebe |

Figure 5: Visual illustrations showcasing various class examples from the benchmarking datasets. Under each image, we provide the corresponding species scientific name (top row) and English common name (bottom row). (a) iNat encompasses a broad spectrum of species, including mammals, plants, and fungi. (b) Aves comprises 200 bird species extracted from the iNaturalist2018 dataset. (c) Flowers102 is a collection of 102 distinct flower species. (d) CUB200 is a bird species recognition dataset that includes 200 bird species.

Table 3: We show the the counts of species scientific and common names which have occurance in LAION400m (Schuhmann et al., 2021), which is the training set of the VLM model OpenCLIP used in our work. We find that a substantial portion of species scientific names (especially from iNat) are not in LAION400M. This helps explain why using scientific names in prompts does not work well for zero-shot species recognition. Moreover, when also counting common names altogether with scientific names, we find that almost all species have then names in LAION400M. This explains why our F-name (using more frequent names of each species in prompts) generally performs the best. It is worth noting that in Flowers102, nearly all scientific names are in LAION400M, explaining why the vanilla prompt method using species scientific names performs exceptionally well on this dataset (Table 2).

| Names Found | iNat (810) | Aves (200) | Flowers102 (102) | CUB200 (200) |
|---|---|---|---|---|
| S-names | 468 | 173 | 98 | 190 |
| + C-names | 781 | 200 | 101 | 200 |

# Appendix

## A Visual Examples

We conducted experiments on four distinct datasets, each primarily focusing on a unique set of species. Aves (Su and Maji, 2021a) and CUB200 (Wah et al., 2011) encompass 200 bird species each. iNat (Su and Maji, 2021b) consists of a diverse range of species, including plants, animals, and fungi, totaling 810 species. Flowers102 (Nilsback and Zisserman, 2008) is a relatively smaller dataset specifically dedicated to 102 flower types. Fig. 5 displays some random images from these datasets.

## B Names Covered by Pretraining Data

We use the VLM model of OpenCLIP (Ilharco et al., 2021) in our work, and the training set is

Table 4: We show top-1 accuracies (in %) by S-name (using scientific names in prompts) and C-name (using common names in prompts) methods on different species types of the iNat datasets. This table supplements Fig. 4. Using scientific names improves accuracy on all species types except plants, on which it achieved decreased accuracy 9.4%, compared to using common names (10.83%).

| Species Type | S-name | C-name |
|---|---|---|
| mammals (14) | 20.00 | 61.43 |
| birds (59) | 6.10 | 37.62 |
| fish (13) | 6.15 | 32.30 |
| reptiles (19) | 4.21 | 24.21 |
| snails (8) | 0.00 | 12.5 |
| shellfish (12) | 16.67 | 28.34 |
| amphibians (9) | 4.45 | 15.56 |
| arachnids (28) | 2.14 | 9.29 |
| insects (264) | 2.88 | 9.84 |
| sea urchin (3) | 6.67 | 13.3 |
| fungi (45) | 4.45 | 10.67 |
| plants (336) | 10.83 | 9.40 |

Table 5: We show our examples using Pica pica, a species in the Aves dataset, known as the common magpie. We use the scientific name, Pica Pica, as part of the prompt for the first method (s-name), and the common name, Common Magpie, for the second method (c-name). The frequency of Common Magpie and Pica Pica in LAION400M (Schuhmann et al., 2021) is then examined. Notably, the scientific name, Pica Pica, has a higher frequency and is employed for the third method (f-name). Finally, descriptions for this class are obtained by prompting GPT4, forming our description-based methods as in (Menon and Vondrick, 2022), and use it with scientific, common, and high-frequency names.

| Prompt Method | Prompt Example |
|---|---|
| S-name | Here is a photo of the Pica pica. |
| + descriptions | + Pica pica has *a blue tail*. |
| C-name | Here is a photo of the common magpie. |
| + descriptions | + Common magpie has *a blue tail*. |
| F-name | Here is a photo of the common magpie. |
| + descriptions | + Pica pica has *a blue tail*. |

LAION400M (Schuhmann et al., 2021). Table 3 demonstrates that scientific names alone have limited occurrence in LAION400M, particularly in more specialized datasets like iNat and Aves. For a comprehensive analysis, please refer to the caption.

## C Breakdown Accuracy for Different Species Types

Table 4 lists the accuracy of each species type on the iNat dataset by prompting s-name and c-name, respectively. It augments Fig. 4.

## D Examples of Prompts

Table 5 shows some prompt examples. Vanilla prompts employs the prompt template "*Here is a photo of the* species-scientific-name". When additionally using descriptions, we adopt the same template as in (Menon and Vondrick, 2022), which is "species-scientific/common-name has blah-blah", where blah-blah can be shape (e.g., "a long tail"), color (e.g., "a blue tail"), etc.

## Limitations

Vision-language models (VLMs) and large-language models (LLMs) could learn bias and unfairness from their pretraining set. As our work exploits publicly available VLMs and LLMs, we do not address these issues and do not study how zero-shot species recognition suffers from them. Moreover, our work focuses on zero-shot setup for species recognition. In practice, even though species-level annotations demand expertise knowledge and are expensive to obtain, there are some annotated data in online libraries, online collections and museums. Leveraging such available data to improve species recognition is future work.

## Ethics Statement

We focus on recognizing species of organisms (e.g., animals, birds, fungi, plants, etc.), we do not envision ethic issues in our work.

## Acknowledgements

This work is supported by NSFC (No.62206256). Shu Kong acknowledges the support by the University of Macau (SRG2023-00044-FST). Shubham Parashar acknowledges the compute resource and travel support from the Department of Computer Science and Engineering at Texas A&M University.

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
