# OpenReview forum: "Prompting Scientific Names for Zero-Shot Species Recognition"
_EMNLP/2023/Conference — EMNLP 2023 Main_

### Official Review · Reviewer_ik9i · 2023-08-04

**Soundness:** 3

**Excitement:**

3: Ambivalent: It has merits (e.g., it reports state-of-the-art results, the idea is nice), but there are key weaknesses (e.g., it describes incremental work), and it can significantly benefit from another round of revision. However, I won't object to accepting it if my co-reviewers champion it.

**Paper Topic And Main Contributions:**

This paper proposes a simple method to convert scientific names into universal names during model inference, which improves the accuracy of zero sample species recognition based on ViT by 2-5 times.

**Reasons To Accept:**

This paper presents a method to facilitate the conversion of scientific names into universal names  for the purpose of enhancing the accuracy of zero sample species recognition using Vision Transformer (ViT) models.

The proposed method is relatively simple yet effective.

**Reasons To Reject:**

While the researchers implemented a method for translating text during the model inference stage, the proposed approach is relatively simplistic and does not appear to introduce any new or innovative techniques or algorithms.

Additionally, the research fails to conduct a horizontal comparison of the prompt template, which could provide valuable insights into the efficacy and accuracy of the underlying model.

**Reproducibility:**

4: Could mostly reproduce the results, but there may be some variation because of sample variance or minor variations in their interpretation of the protocol or method.

**Reviewer Confidence:**

4: Quite sure. I tried to check the important points carefully. It's unlikely, though conceivable, that I missed something that should affect my ratings.

---

> ### Author Rebuttal · Authors · 2023-08-28
>
> We thank Reviewer ik9i for the valuable comments and suggestions. Reviewer ik9i appreciates our method and thinks it is "simple yet effective". Reviewer ik9i has two concerns and we address them below.
>
> > **Reviewer ik9i worries that the method is too simple and our work "does not appear to introduce any new or innovative techniques or algorithms".**
>
> Perhaps Reviewer ik9i finds the simplicity of our method to prevail the novelty of our work. The literature shows that simple methods tend to make big impacts.
>
> To the best of our knowledge, our work is the first that shows improving the vocabulary of category names (i.e., translating scientific names to common names) significantly improves a downstream task (i.e., zero-shot species recognition). In contrast, prior work has focused on more sophisticated techniques to exploit attributes [1] and prompt templates [2] to improve zero-shot performance. Our simple method is effective (as noted by reviewers). For example, it achieves 39.8% accuracy on the Aves dataset (Table 1), significantly higher than 7.05% by [1] and 7.75% by [2]! Surprisingly, our zero-shot approach even surpasses the fully supervised learning method which achieves 20.6% by [3]! Our finding reveals the significant but overlooked insight that, *aligning a downstream task's vocabulary of categorical names to the more frequently seen (scientific/common) names by the pretrained model OpenCLIP allows leveraging this model to better serve the downstream task*. Therefore, we believe our work is novel and invaluable to our community!
>
>  - [1] Radford et al., "Learning Transferable Visual Models From Natural Language Supervision". ICML, 2021.
>  - [2] Menon et al., "Visual classification via description from large language models". ICLR, 2023.
>  - [3] Su, et al., "A Realistic Evaluation of Semi-Supervised Learning for Fine-Grained Classification". CVPR, 2021
>
>
>
> > **”Reviewer ik9i suggests "a horizontal comparison of the prompt template, which could provide valuable insights into the efficacy and accuracy of the underlying model."**
>
> Great suggestion! Below, we study the zero-shot performance using different prompt templates as suggested by [1]. Clearly, our method still performs the best across prompt templates, and results show quite small deviations. That said, our study is not affected by the choice of prompt templates. We will add them to the appendix of our camera-ready paper. Thank you for the suggestion!
>
> | Prompting Method          | iNat | Aves | Flowers102 | CUB200 |
> |-----------------| ------- | ------ |-------- | ------ |
> | `itap of a {}`              |                    |                    |                     |                 |
> |                    + s-name |                6.2 |                7.1 |                67.0 |            5.7 |
> |                    + c-name |               13.2 |               35.9 |                64.6 |           54.3 |
> |                    + f-name (**ours**) |               13.1 |               36.2 |                69.4 |           55.2 |
> | `a bad photo of the {}`     |                    |                    |                     |                 |
> |                    + s-name |                6.3 |                7.6 |                65.9 |            5.8 |
> |                    + c-name |               13.6 |               39.6 |                64.3 |           56.3 |
> |                    + f-name (**ours**)  |               13.9 |               39.5 |                68.7 |           57.1 |
> | `a photo of the large {}`   |                    |                    |                     |                 |
> |                    + s-name |                6.4 |                7.4 |                66.5 |            6.3 |
> |                    + c-name |               13.6 |               38.0 |                63.5 |           56.3 |
> |                    + f-name (**ours**)  |               13.9 |               38.0 |                68.2 |           57.6 |
> | `a photo of the small {}`   |                    |                    |                     |                 |
> |                    + s-name |                6.6 |                6.8 |                66.0 |            6.2 |
> |                    + c-name |               13.3 |               38.4 |                63.4 |           56.3 |
> |                    + f-name (**ours**)  |               13.7 |               38.3 |                68.5 |           57.1 |
> | `a {} in the wild`         |                    |                    |                     |                 |
> |                    + s-name |                6.2 |                6.6 |                67.3 |            5.9 |
> |                    + c-name |               13.6 |               39.4 |                63.7 |           58.5 |
> |                    + f-name (**ours**)  |               13.8 |               39.6 |                68.8 |           59.2 |
> | `mean acc w/ std`          |                    |                    |                     |                 |
> |                    + s-name |6.35 &plusmn; 0.15    |7.1&plusmn;0.37 |66.51&plusmn;0.60|5.97&plusmn;0.27 |
> |                    + c-name |13.46 &plusmn; 0.17 |38.25&plusmn;1.33|63.92&plusmn;0.46 |56.4&plusmn;1.35|
> |                    + f-name  (**ours**) |13.68 &plusmn; 0.29 |38.29&plusmn;1.40|68.74&plusmn;0.44|57.23&plusmn;1.44|
>
>
>  - [1] Radford et al., "Learning Transferable Visual Models From Natural Language Supervision". ICML, 2021.

---

### Official Review · Reviewer_tzwz · 2023-08-05

**Typos Grammar Style And Presentation Improvements:** The paper is well-written and easy to…
**Soundness:** 4

**Excitement:**

3: Ambivalent: It has merits (e.g., it reports state-of-the-art results, the idea is nice), but there are key weaknesses (e.g., it describes incremental work), and it can significantly benefit from another round of revision. However, I won't object to accepting it if my co-reviewers champion it.

**Missing References:**

a few interesting papers to add for the ZSL part.

Karessli N. et al. (2016) Gaze Embeddings for Zero-Shot Image Classification. In CVPR.
Badirli et. al. (2023) Classifying the unknown: Insect identification with deep hierarchical Bayesian learning. Methods in Ecology and Evolution.

**Paper Topic And Main Contributions:**

The paper proposes a simple yet effective idea to use VLMs for zero-shot image classification, in particular for fine-grained datasets.

The main contribution of the method is the idea of translating the Latin names in fine-grained datasets to common language to help VLMs' language model to better accommodate the ZSL task.

**Questions For The Authors:**

A: Considering most of the class names from CUB were in the CLIP pre-training dataset (Table 3), can authors elaborate why there is a massive performance boost with the proposed method?

**Reasons To Accept:**

The proposed idea is very simple and it is validated on 4 benchmark datasets by consistently outperforming the baselines.

The idea of using VLMs for fine-grained datasets is a bold move and can open-up an interesting research venue.


**Reasons To Reject:**

In real-world scenario with hundreds of species from the same genus or family, I wonder how this method will generalize. To illustrate, there are more than 2000 beetles and not many of them has common names. One of the reasons that the model works well on CUB dataset is that the CUB dataset is not very fine-grained. Most of bird species are well known ones and have common names. There are most 4-5 species belonging to the same genus in that dataset.

No ZSL methods are tested as a baseline. It would be very beneficial for the paper to show how the proposed method (VLM + translated class names) performs against some of traditional ZSL methods [1, 2, 3, 4]



[1] Xian, Y., Lampert, C. H., Schiele, B., & Akata, Z. (2018). Zero- shot learning— A comprehensive evaluation of the good, the bad and the ugly. IEEE Transactions on Pattern Analysis and Machine Intelligence, 41(9), 2251– 2265.
[2] E. Schonfeld, S. Ebrahimi, S. Sinha, T. Darrel, and Z. Akata. Generalized zero- and few-shot
learning via aligned variational autoencoders. In CVPR, 2019.
[3] S Badirli, Z Akata, G Mohler, C Picard, M Dundar. Fine-Grained Zero-Shot Learning with DNA as Side Information. In NeurIPS, 2021.
[4] B. Romera-Paredes and P. H. Torr. An embarrassingly simple approach to zero-shot learning.In ICML, 2015.

**Reproducibility:**

4: Could mostly reproduce the results, but there may be some variation because of sample variance or minor variations in their interpretation of the protocol or method.

**Reviewer Confidence:**

5: Positive that my evaluation is correct. I read the paper very carefully and I am very familiar with related work.

---

> ### Author Rebuttal · Authors · 2023-08-28
>
> We thank Reviewer tzwz for the valuable comments. Reviewer tzwz thinks the "proposed idea is very simple" which "is a bold move and can open-up an interesting research venue." The reviewer has three concerns and we address them below.
>
> > **Reviewer tzwz questions whether our method can generalize because, as stated by the reviewer, "one of the reasons that the model works well on CUB dataset is that the CUB dataset is not very fine-grained" (which has only 200 species, and each genus has <5 species), whereas "there are more than 2000 beetles".**
>
> The statistics of the CUB dataset is correct, but we respectfully note that the CUB dataset is well-established in the field of fine-grained recognition although it has <5 species per genus. Importantly, another dataset we focus on (Table 1, Fig. 3-4) is iNat, which is more challenging than CUB because it has 810 species and some genera have upto 10 species. As noted by the reviewer, we evaluate our method "on 4 benchmark datasets"; extensive results sufficiently demonstrate that our method generalizes well across datasets. We agree that nature has way more species (e.g., >2000 beetles), but there does not exist such large-vocabulary fine-grained datasets for experiments. We hope future work will collect such a dataset because we are also curious about how typical zero-shot recognition methods and ours perform on it!
>
>
> > **Reviewer tzwz comments that "No ZSL methods are tested as a baseline" and recommends some ZSL papers.**
>
> Thank you for the recommended papers and we will happily cite them in the camera-ready. We would like to first emphasize the crucial difference (as briefly contrasted in Line 096-112) between the classic zero-shot learning (ZSL) setup and the current zero-shot recognition setup enabled by pretrained vision-language models (VLMs).
>
>  - The classic ZSL setup in computer vision refers to the study of generalizing to **unseen classes** in image classification while performing knowledge transfer from seen classes using some semantic information such as attributes. The papers you recommend fall in this setting.
>
>  - The setup in our work refers to the new **zero-shot transfer** setup formulated in the CLIP paper [1] that aims to leverage a pre-trained VLM for zero-shot recognition in **unseen imagery datasets** prompted by texts, i.e., allowing methods to be based on prompting.
>
> In the new setup, [1] shows the prompting-based approach that uses pre-trained VLMs resoundingly outperforms classic ZSL methods. Therefore, we treat the suggested prompting method [1] as our strong baseline. Furthermore, we compare the most recent prompting method [2] which uses LLMs (i.e., GPT3) to extract more semantic information to augment text prompts for better zero-shot performance. In our work, we reimplement [2] by using a more powerful LLM (i.e., GPT-4) to report its performance (Table 1). We will clarify and discuss the above in the camera-ready version.
>
>
>  - [1] Radford et al., "Learning Transferable Visual Models From Natural Language Supervision". ICML, 2021.
>  - [2] Menon et al., "Visual classification via description from large language models". ICLR, 2023.
>
>
>
>
> > **Reviewer tzwz asks, "considering most of the class names from CUB were in the CLIP pre-training dataset (Table 3), can authors elaborate why there is a massive performance boost with the proposed method" (which incorporates common names and use the most frequent name for a given species in prompting).**
>
> Great question! The conclusive explanation is that, *aligning a downstream task's vocabulary of categorical names to the more frequently seen (scientific/common) names by the pretrained model OpenCLIP allows leveraging this model to better serve the downstream task*. Let us explain further below.
>
> Intuitively, when a pretrained model has seen common names much more frequently than scientific names, it understands the common names better. To justify this, we assemble results on the CUB dataset in the table below. The table lists the counts of scientific and common names included in OpenCLIP's pretraining dataset LAION400M, and zero-shot accuracies by prompting the two sets of names, respectively. Clearly, common names have been seen by OpenCLIP more than 10 times than scientific names, and prompting common names yields 56.01% zero-shot accuracy, 9x better than prompting scientific names (6.08%)!
>
> | #captions with that have | s-names | c-names |
> |------|------:|----:|
> | counts |  49,506     | 621,275     |
> | zero-shot accuracy using  | 6.08%  | 56.01% |
>
> Lastly, our final method is to prompt more frequent names among scientific and common names, hence we call this method f-name. From Table 2 of our paper, simply doing so improves accuracy to 58.35%, notably higher than using common names only (56.01% as shown by the table above). This further demonstrates the conclusion that, *aligning a downstream task's vocabulary of categorical names to the more frequently seen (scientific/common) names by the pretrained model OpenCLIP allows leveraging this model to better serve the downstream task*.

---

### Official Review · Reviewer_Duti · 2023-08-06

**Typos Grammar Style And Presentation Improvements:** No typos that I could see.
**Soundness:** 4

**Excitement:**

4: Strong: This paper deepens the understanding of some phenomenon or lowers the barriers to an existing research direction.

**Missing References:**

No missing reference to my knowledge.

**Paper Topic And Main Contributions:**

This paper describes a way to improve retrieval of bird, plant and animal species using scientific names through a translation approach in which they translate the scientific name into a common name before prompting a large multimodal model such as CLIP. The authors achieve retrieval results that exceed the state of the art by an order of magnitude.

**Questions For The Authors:**

Have you considered the lack of a one one onto mapping between common names and scientific names? How does your method deal with ambiguities between a horse and a sea horse or a lion and a sea lion or  a pig and a guinea pig?
Update: the authors have provided a convincing response.

**Reasons To Accept:**

1. Interesting finding backed by great improvement over the state of the art.

**Reasons To Reject:**

1. The technique itself is "embarrasingly" simple as per the authors own description. While that is not a knock against it, the authors do not dig deeper into what the implications of their result are. It is not shocking that CLIP would be trained with data that does not have scientific names in it. However, scientific names are assigned so there are unambiguous names for species. Common names on the other hand are often ambiguous and downright confusing. For example, prairie dogs are rodents while dogs are canines, Similarly guinea pigs are not porcine at all.  A lion is a feline but a sea lion is a mariine mammal. The authors are satisfied with the improvements in the results that they have achieved but do not probe for cases where there may not be a bijective mapping between scientific and common names.
update: The authors have provided a convincing response to this question.

**Reproducibility:**

5: Could easily reproduce the results.

**Reviewer Confidence:**

5: Positive that my evaluation is correct. I read the paper very carefully and I am very familiar with related work.

---

> ### Author Rebuttal · Authors · 2023-08-28
>
> We thank Reviewer Duti for the insightful comments. Reviewer Duti thinks our work has an "interesting finding backed by great improvement over the state of the art." Reviewer Duti has a major concern about ambiguity in common names, as well as a minor one about our "embarrassingly simple" method. We address them below.
>
> > **Reviewer Duti thinks "common names on the other hand are often ambiguous and downright confusing" because "there may not be a bijective mapping" between scientific names and common names. The reviewer explains this ambiguity with multiple examples and asks how we handle this issue, e.g., "prairie dogs" vs. "dogs",  "guinea pigs" vs. "pigs", "a lion" vs. "a sea lion", "a horse" vs. "a seahorse".**
>
> Great comment! The comment is partially correct -- it is right that there might not always be one-to-one (bijective) mapping. Concretely, it is quite often that a scientific name can have multiple common names. But, importantly, a common name corresponds to only one scientific name. To quickly justify this, we can find the common names for a scientific name using wikipedia; for a common name, we can do the same but wikipedia "translates" the common name to a unique scientific name. That said, common names are not ambiguous. We explain further below.
>
>
> The examples provided by the reviewer look ambiguous to a human because they have the same keywords, e.g., keyword "lion" in "a lion (Panthera Leo)" vs. "a sea lion (Otariinae)", keyword "horse" in "a horse (Equus caballus)" vs. "a seahorse (Hippocampus guttulatus)".
> For the examples provided by the reviewers, we find their corresponding scientific names using wikipedia, shown below. Clearly, each common name corresponds to a single unique scientific name. Hence, common names are not ambiguous.
>  - lion → Panthera Leo
>  - sea lion → Otariinae
>  - horse → Equus caballus
>  - seahorse → Hippocampus guttulatus
>  - pigs →  S. domesticus
> - guinea pigs → Cynomys socialis
>  - prairie dogs → Cynomys socialis
>  - dogs → C. familiaris
>
> Importantly, pre-trained vision-language models (e.g., OpenCLIP) have learned *the correspondence between (unambiguous) common names and visual characteristics*, hence using common names (frequently seen in the pretraining set) for zero-shot recognition is much better than using infrequently-seen scientific names.  To support our claim, we did two experiments of binary zero-shot recognition  between (1) lion vs. sea lion, and (2) pigs vs. guinea pigs. These classes are in the popular Imagenet-1K dataset and we use OpenCLIP to zero-shot recognize their images (following the standard text-prompt method described in our paper). Below are the confusion matrices between the two pairs of classes. Clearly, OpenCLIP does not confuse these common classes!
>
> - Confusion matrix of zero-shot recognition between lion and sea lion.
>   |class| lion| sea  lion|
>   |--------|---------:|------:|
>   |lion| 100.0% |  0.0%|
>   |sea lion       | 0.0%  | 100.0%|
>
> - Confusion matrix of zero-shot recognition between pig vs guinea pig.
>   |class | pig| guinea pig|
>   |--------|---------:|------:|
>   |pig     | 100.0% |  0.0%|
>   |guinea pig| 6.0%  | 94.0%|
>
>
> > **Reviewer Duti is concerned that our method is “embarrassingly simple” and suggests doing some analysis to understand "what the implications of their result are".**
>
> In the literature, simple methods tend to make big impacts. To the best of our knowledge, our work is the first that shows improving the vocabulary of category names (i.e., translating scientific names to common names) significantly improves a downstream task (i.e., zero-shot species recognition). In contrast, prior work has focused on more sophisticated techniques to exploit attributes [1] and prompt templates [2] to improve zero-shot performance. Hence, our method is not only simple but also novel in this field.
>
> In terms of more analyses, we have listed some in Table 3 in the appendix after reference. May we politely ask the reviewer to have a look and we are delighted to do more per your suggestion. Yet, towards a four-page short paper, it is difficult to include all the analyses in the main text. The implication of our results enables rich future work as briefly mentioned in Line274-280, e.g., how to leverage the frequently-seen common names to finetune / adapt such pre-trained models in downstream tasks.
>
>  - [1] Menon et al., "Visual classification via description from large language models", ICLR, 2023.
>  - [2] Radford et al., "Learning Transferable Visual Models From Natural Language Supervision". ICML, 2021.

---

### Meta-Review · Area_Chair_bYnn · 2023-09-09

**Recommendation:** 5

**Metareview:**

The paper proposes a neat yet effective promoting method for scientific name understanding in VLMs, which is a good example of a short paper. The reviewing process is generally towards positive and the rebuttal helps to answer some of the reviewers' concern. Please help include the additional results in any final versions.

---

### Decision · Program_Chairs · 2023-10-07

**Decision:**

Accept-Main

**Comment:**

The paper proposes a neat yet effective promoting method for scientific name understanding in VLMs, which is a good example of a short paper. The reviewing process is generally towards positive and the rebuttal helps to answer some of the reviewers' concern. Please help include the additional results in any final versions.